# Developing and validating an explainable digital mortality prediction tool for extremely preterm infants

T'ng Chang Kwok[1,2], Chao Chen[3], Jayaprakash Veeravalli[3], Carol A.C. Coupland[4], Edmund Juszczak[5], Jonathan Garibaldi[3,6], Kirsten Mitchell[7], Kate L. Francis[8,9], Christopher J. D. McKinlay[10,11], Brett J. Manley[8,12,13], Don Sharkey[1]*

**1** Centre for Perinatal Research, Lifespan and Population Health, School of Medicine, University of Nottingham, Nottingham, United Kingdom, **2** Nottingham Neonatal Service, Nottingham Univesity Hospitals NHS Trust, Nottingham, United Kingdom, **3** School of Computer Science, University of Nottingham, Nottingham, United Kingdom, **4** Centre for Academic Primary Care, Lifespan and Population Health, School of Medicine, University of Nottingham, Nottingham, United Kingdom, **5** Nottingham Clinical Trials Unit, School of Medicine, University of Nottingham, Nottingham, United Kingdom, **6** Provost Office, University of Nottingham, Ningbo, China, **7** Spoons Charity, Manchester, United Kingdom, **8** Murdoch Children's Research Institute, Melbourne, Victoria, Australia, **9** Department of Paediatrics, The University of Melbourne, Melbourne, Victoria, Australia, **10** Kidz First Neonatal Care, Te Whatu Ora Counties Manukau, Auckland, New Zealand, **11** Department of Paediatrics, Child and Youth Health, University of Auckland, Auckland, New Zealand, **12** Newborn Research, The Royal Women's Hospital, Melbourne, Victoria, Australia, **13** Department of Obstetrics, Gynaecology and Newborn Health, The University of Melbourne, Melbourne, Victoria, Australia

\* don.sharkey@nottingham.ac.uk

## Abstract

Decision-making in perinatal management of extremely preterm infants is challenging. Mortality prediction tools may support decision-making. We used population-based routinely entered electronic patient record data from 25,902 infants born between $23^{+0}$–$27^{+6}$ weeks' gestation and admitted to 185 English and Welsh neonatal units from 2010–2020 to develop and internally validate an online tool to predict mortality before neonatal discharge. Comparing nine machine learning approaches, we developed an explainable tool based on stepwise backward logistic regression (https://premoutcome.shinyapps.io/Death/). The tool demonstrated good discrimination (area under the receiver operating characteristics curve (95% confidence interval) of 0.746 (0.729–0.762)) and calibration with superior net benefit across probability thresholds of 10%–70%. Our tool also demonstrated superior calibration and utility performance than previously published models. Acceptable performance was demonstrated in a multinational, external validation cohort of preterm infants. This tool may be useful to support high-risk perinatal decision-making following further evaluation.

**Data availability statement:** Ethical restrictions apply to the availability of the potentially identifying and sensitive patient data used in the study, and so are not publicly available. Professor Don Sharkey had full access to the NNRD data in the study. However, the NNRD data that support the findings of this study are available from the Neonatal Data Analysis Unit (https://healthdatagateway.org/en/dataset/619 or email: ndau@imperial.ac.uk) upon appropriate ethics approval. Professor Brett Manley and Dr Kate Francis had full access to the PLUSS trial data, which is available from the PLUSS trial study team (https://www.plusstrial.org/contact/ or email: pluss@thewomens.org.au) upon appropriate ethics approval.

**Funding:** TCK received the Action Medical Research training fellowship (https://action.org.uk/) supported by the Albert Gubay Foundation, as part of this study. DS was funded by the National Institute for Health and Care Research (NIHR) Children and Young People MedTech Co-operative (CYP MedTech) (https://hrc-children.nihr.ac.uk/). BJM was funded by the National Health and Medical Research Council (Australia) (https://www.nhmrc.gov.au/). The views expressed are those of the author(s) and not necessarily those of the NHS, the NIHR or the Department of Health and Social Care. The funders play no role in the study design, data collection and analysis, decision to publish, or preparation of the manuscript.

**Competing interests:** The authors have declared that no competing interests exist.

## Author summary

Increasingly, more premature babies are being born even earlier and surviving. Each premature baby is unique, with different combinations of factors affecting their chances of survival. An individualised approach is needed to support discussions with parents in creating a care plan for the baby before birth. Prediction tools can help support this discussion and reduce variation in the care delivered by providing an objective measure after considering important risk factors. We used artificial intelligence to analyse the electronic health records of 25,902 premature babies born between 23 and 27 completed weeks of pregnancy from 2010 to 2020 in England and Wales. We worked with parent groups to use the data pattern identified by artificial intelligence to develop an online tool (https://premoutcome.shinyapps.io/Death/) to predict the risk of premature babies dying. The tool demonstrated how the risk factors contributed to the prediction, explaining how the predicted risk was derived. The tool developed demonstrated better performance than previously developed tools in our cohort of babies in England and Wales. The tool also showed good performance when tested in a separate cohort of babies in Australasia. The tool developed could support parental discussion and decision-making following further evaluation.

## Introduction

Preterm births are increasing in industrialised countries and are the leading cause of mortality in children under five [1]. Over the last 20 years, survival of extremely preterm infants born before 28 weeks' gestation has improved across the UK [2,3] and other developed countries [4,5]. Approximately 50% of neonatal unit deaths of extreme preterm infants occur in the first week of life [3]. Parental counselling and preterm birth decision-making surrounding the perinatal management of infants born at the extremes of prematurity can be very challenging [6]. Parents prefer counselling to be personalised, to balance hope and realistic expectations, and to offer shared decision-making [7]. Perinatal clinical teams may lack recent risk-based data to allow more individualised counselling and decision-making, potentially introducing bias and personal experience into the discussion, resulting in variability with subsequent care of the infant.

Internationally, there is marked variation in active treatment of extremely preterm infants, especially those born less than 25 weeks' gestation, which can be dependent on the gestational days of the pregnancy [8] and can influence care pathways with variation with in-utero transfer to a tertiary care centre or initiation of antenatal corticosteroids [9–11]. Difficult decisions around active treatment or the need to consider transfer could be influenced by outcome data, which will differ based on the characteristics of the pregnancy.

To try and address these issues, national bodies have written guidance [12,13] on the perinatal management of extremely preterm infants with emphasis on two key

areas. Firstly, the need for an individualised approach in decision-making, taking into account multiple factors, moving away from gestational age or birth weight cut-offs alone. Secondly, the importance of supporting parental involvement in decision-making by providing clear antenatal information.

The current recommended risk-based approach to perinatal decision-making [14] is still subjective and open to misinterpretation. Although risk factors are listed in the recommendations [12,13], it is unclear how these factors interact with one another to determine infant outcomes. Hence, a multivariable prediction tool combining key perinatal characteristics could provide an objective measure to support complex shared decision-making. Although several mortality prediction models have been developed [2,15–17], none are used in routine clinical practice to predict individual outcomes or provide users with estimates of the importance of characteristics for that infant.

Using population-based, routinely entered electronic patient record data in England and Wales and nine machine learning approaches, we aimed to develop and internally validate a mortality prediction tool for extremely preterm infants. We planned to compare our developed prediction tool with previously published models [2,15–17] in a separate 'test' cohort, and externally validated the tool in an international cohort of preterm infants.

## Results

### Mortality prediction tool development

**Study population.** A total of 25,902 infants born between $23^{+0}$ and $27^{+6}$ weeks' gestation were included in the study cohorts to develop and internally validate the prediction tool. Of these, 5,550 (21%) infants died before neonatal discharge. 1,642 (6%) infants with missing data for the nine predictors used in the modelling were excluded from the analysis (S1 Table). The proportion of deaths in infants in the 'test' cohort was lower than that in the 'training and validation' cohort (20% vs 22%). The infants in the 'test' cohort were also slightly more preterm and had a lower median birth weight z score. Mothers of infants in the 'test' cohort were more likely to have received a complete course of antenatal corticosteroids (ANC) and delivered in a centre with a co-located neonatal intensive care unit (NICU) (Table 1).

**Algorithm development.** The nine machine learning approaches are described in S1 File. Across all machine learning approaches used, gestational age and birth weight z-score were consistently the two most important predictors based on the mean SHapley Additive exPlanations (SHAP) values (Fig 1). Chorioamnionitis and congenital anomalies were the least important predictors (S1 Fig, S2 Table). Most of the predicted risks derived from the nine machine learning approaches were below 20% (S3 Table).

**Internal validation (discrimination, calibration and utility).** Discrimination (Area under the receiver operating characteristics curve): There was little difference in the discrimination performance in the 'test' cohort in terms of area under the receiver operating characteristics curve (AUROC) between the seven approaches of extreme gradient boosting, feedforward neural network, random forest, long short-term memory, adaptive neuro-fuzzy inference system, AutoPrognosis 2.0, and logistic regression with AUROCs ranging from 0.746 to 0.757 (Table 2).

Calibration (Calibration plot, calibration-in-the-large and calibration slope): Extreme gradient boosting, feedforward neural network, long short-term memory, and logistic regression had good calibration across the range of predicted mortality risk (Table 2, Fig 2).

Utility (Decision curve analysis): All the approaches, except for support vector machine, also had a similar superior net benefit compared with providing palliative/comfort care for all or no infants across a reasonable range of mortality threshold probabilities between 10% to 70% when deciding perinatal management in the decision curve analysis (Fig 3).

The logistic regression model (S4 Table) was further developed in preference to the other machine learning approaches due to its familiarity among healthcare professionals and ease of interpretation, while achieving similar model performance in the 'test' cohort.

Table 1. Characteristics of infants in the cohorts used to develop, internally and externally validate the prediction model.

| Clinical characteristics | Development and internal validation cohorts | | 'External validation' cohort (N = 1,052) |
| --- | --- | --- | --- |
| | 'Training & Validation' cohort (N = 19,406) | 'Test' cohort (N = 6,496) | |
| **Gestation at birth** (weeks), median (IQR) | $26^{+1}$ $(24^{+6}–27^{+1})$ | $26^{+0}$ $(24^{+6}–27^{+0})$ | $25^{+4}$ $(24^{+5}–26^{+5})$ |
| **Birth weight Z score**[1], median (IQR) | -0.37 (-0.88 to 0.09) | -0.41 (-0.94 to 0.07) | -0.12 (-0.85 to 0.48) |
| Missing, n (%) | 0 (0) | 1 (0.02) | 0 (0) |
| **Sex**, n (%) | | | |
| Male | 10,557 (54) | 3,525 (54) | 582 (55) |
| Missing | 0 (0) | 1 (0.02) | 0 (0) |
| **Multiple pregnancy**, n (%) | 4,862 (25) | 1,404 (22) | 295 (28) |
| Missing | 2 (0.01) | 1 (0.02) | 0 (0) |
| **Antenatal corticosteroids**[1], n (%) | | | |
| Complete course | 12,688 (65) | 4,542 (70) | 700 (67) |
| Incomplete course | 3,813 (20) | 1,334 (21) | 309 (29) |
| No course | 1,882 (10) | 5 (0.08) | 43 (4) |
| Missing | 1,023 (5) | 615 (9) | 0 (0) |
| **Congenital anomaly**, n (%) | 461 (2) | 239 (4) | 0 |
| **Chorioamnionitis**, n (%) | 1,464 (8) | 516 (8) | 480 (46) |
| Missing | 0 (0) | 0 (0) | 1 (0.1) |
| **Prolonged rupture of membranes**, n (%) | 2,688 (14) | 688 (11) | 324 (31) |
| Missing | 0 (0) | 0 (0) | 1 (0.1) |
| **Born in a centre with NICU**, n (%) | 13,139 (68) | 4,880 (75) | 1,017 (97) |
| Missing | 3 (0.02) | 0 (0) | 0 (0) |
| **Maternal ethnicity**, n (%) | | | |
| White | 12,028 (62) | 3,275 (50) | 611 (58) |
| South Asian | 2,256 (12) | 742 (11) | N/A |
| Black | 2,113 (11) | 581 (9) | N/A |
| Aboriginal or Torres Strait Islander | N/A | N/A | 57 (5) |
| African or Indigenous | N/A | N/A | 29 (3) |
| Asian | N/A | N/A | 208 (20) |
| First Nations, Ink/Inuit or Metis | N/A | N/A | 13 (1) |
| Hispanic | N/A | N/A | 10 (1) |
| Māori | N/A | N/A | 77 (7) |
| Pacific Islander | N/A | N/A | 42 (4) |
| Others/Mix | 720 (4) | 243 (4) | 5 (0.5) |
| Missing | 2,289 (12) | 1,655 (25) | 0 (0) |
| **Neonatal network at birth**, n (%) | | | |
| Network 1 | 1,075 (6) | 380 (6) | N/A |
| Network 2 | 1,489 (8) | 499 (8) | |
| Network 3 | 1,696 (9) | 517 (8) | |
| Network 4 | 1,062 (5) | 361 (6) | |
| Network 5 | 2,704 (14) | 848 (13) | |
| Network 6 | 991 (5) | 335 (5) | |
| Network 7 | 1,441 (7) | 524 (8) | |
| Network 8 | 1,400 (7) | 458 (7) | |
| Network 9 | 1,205 (6) | 352 (5) | |
| Network 10 | 1,585 (8) | 502 (8) | |

*(Continued)*

 

**Table 1.** (Continued)

| Clinical characteristics | Development and internal validation cohorts | | 'External validation' cohort (N = 1,052) |
| --- | --- | --- | --- |
| | 'Training & Validation' cohort (N = 19,406) | 'Test' cohort (N = 6,496) | |
| Network 11 | 527 (3) | 278 (4) | |
| Network 12 | 2,070 (11) | 727 (11) | |
| Network 13 | 1,718 (9) | 569 (9) | |
| Missing | 443 (2) | 146 (2) | |
| **Death before neonatal discharge**, n (%) | 4,246 (22) | 1,304 (20) | 206 (20) |

IQR denotes interquartile range (25th to 75th percentile). NICU denotes Neonatal Intensive Care Unit.

[1] Complete course of antenatal corticosteroids was defined as at least two doses of corticosteroids given before delivery.

**Comparison with other published models.** Although the discrimination performance of the Manktelow 2013 [17] and Santhakumaran 2018 [2] models (AUROC 0.758 and 0.754, respectively) were similar to our developed logistic regression mortality prediction model (AUROC 0.746) in the 'test' cohort, they had worse calibration performance (Table 2 and Fig 4). Our logistic regression model was also found to have superior net benefit across the range of mortality probability thresholds of 20% to 80% than the four previously published models (Fig 4).

**Subgroup analysis.** Of the included infants, 1,500 (26%) and 55 (1%) in the 'test' cohort had missing data on maternal ethnicity and neonatal network, respectively. The performance of the logistic regression mortality prediction model in infants born to mothers who identified as Black was worse than in infants born to mothers of other ethnicities. The AUROC was 0.713 with a 95% confidence interval (CI) of 0.655 to 0.770 (S5 Table) with an overestimation of mortality risk (S2 Fig) in infants born to mothers who identified as Black. However, the logistic regression model demonstrated superior net benefit across a range of mortality threshold probabilities between 10% to 50% in all four maternal ethnicities (S3 Fig). There was also variation in the logistic regression model performance across the 13 neonatal networks. In networks with lower neonatal mortality risk, such as Networks 3 and 10, an overestimation of neonatal mortality risk was observed (S5 Table and S2 Fig), and the net benefit of using the model was only found across a narrower range of mortality threshold probabilities (S3 Fig).

### External validation

**Study population.** Our logistic regression model was externally validated on the 1,051 infants recruited to the international, multicentre PLUSS trial [19], after excluding seven infants born before 23 weeks' gestation and one infant with missing data on prolonged rupture of membranes and chorioamnionitis. In this PLUSS cohort, 206 (20%) infants died before neonatal discharge. Infant characteristics are reported in Table 1.

**Model performance.** The logistic regression model had similar discrimination in the 'external validation' cohort with an AUROC of 0.728 (95% CI 0.700 to 0.767) with an overestimation of mortality risk (calibration-in-the-large of -0.27 (95% CI -0.52 to -0.01) and calibration slope of 0.94 (95% CI 0.75 to 1.13)) (Fig 5A). The logistic regression model demonstrated superior net benefit across a range of mortality threshold probabilities between 10% to 40% (Fig 5B).

### Online tool

An online prediction tool based on the logistic regression algorithm was developed and can be accessed at https://pre-moutcome.shinyapps.io/Death/ (Fig 6).

### Discussion

We have developed and validated a risk prediction model, and produced an online mortality prediction tool for extremely preterm infants born $23^{+0}$ to $27^{+6}$ weeks' gestation. We used perinatal factors based on the largest and most recent

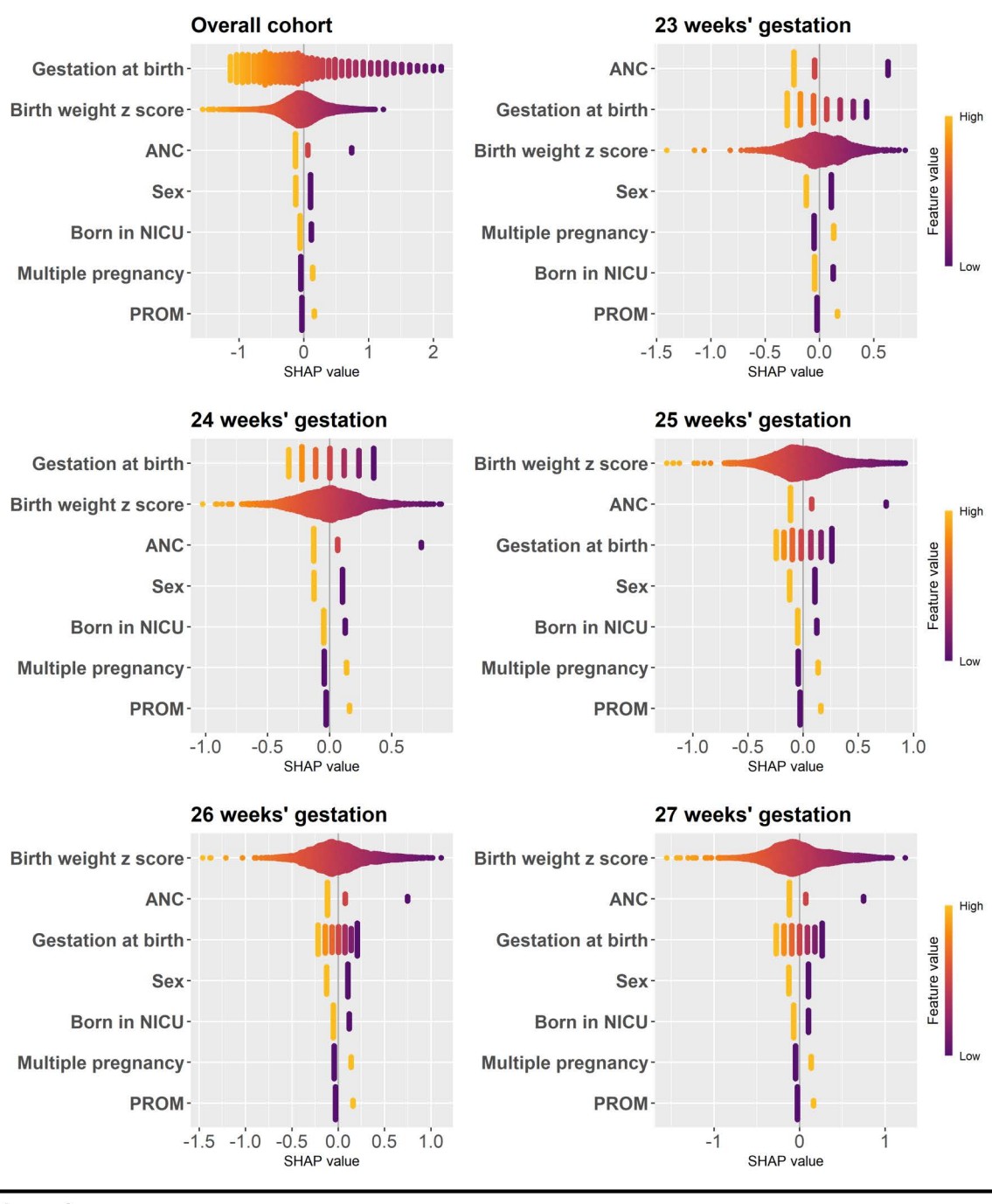

**Legend**
1. A positive SHAP value in the x-axis represents the model prediction closer to death.
2. Gestation at birth denotes the individual gestation day within each gestational week at birth.
3. Female sex was labelled as having a higher value than the male sex.
4. Complete antenatal corticosteroid (ANC) was labelled as having a higher value than incomplete and no ANC.
5. For the predictors "Born in NICU", "Multiple pregnancy" and "Prolonged rupture of membrane (PROM)", "Yes" was labelled as having a higher value than "No".

**Fig 1. Beeswarm plot showing the SHapley Additive exPlanations (SHAP) value of the predictors (ranked in the y-axis by their mean absolute SHAP values) for the overall 'test' cohort and by weeks of gestation based on the logistic regression model.** Chorioamnionitis and congenital anomalies predictors were dropped from the logistic regression model after backward stepwise selection.

**Table 2. Model performance of the machine learning approaches and previously published models in the 'test' cohort (n = 5,879).**

| Algorithm/Model | Discrimination<br>Area under ROC curve | Calibration | |
|---|---|---|---|
| | | Calibration-in-the-large | Calibration slope |
| **Extreme Gradient Boosting** | 0.757 (0.741 to 0.773) | -0.13 (-0.24 to -0.02) | 1.07 (0.99 to 1.15) |
| **Feedforward Neural Network** | 0.757 (0.741 to 0.773) | 0.00 (-0.12 to 0.11) | 1.08 (1.00 to 1.16) |
| **Random Forest** | 0.757 (0.741 to 0.773) | 0.11 (-0.01 to 0.24) | 1.26 (1.17 to 1.35) |
| **Long Short-Term Memory** | 0.755 (0.739 to 0.771) | 0.12 (0.00 to 0.24) | 1.12 (1.03 to 1.20) |
| **Adaptive Neuro-Fuzzy Inference System** | 0.754 (0.738 to 0.771) | -0.26 (-0.36 to -0.15) | 0.95 (0.87 to 1.02) |
| **AutoPrognosis 2.0** | 0.751 (0.735 to 0.768) | 0.00 (-0.11 to 0.12) | 1.10 (1.02 to 1.18) |
| **Logistic Regression** | 0.746 (0.729 to 0.762) | -0.11 (-0.22 to 0.00) | 1.01 (0.94 to 1.09) |
| **K-Nearest Neighbour** | 0.737 (0.720 to 0.754) | -0.38 (-0.50 to -0.27) | 0.72 (0.65 to 0.80) |
| **Support Vector Machine** | 0.718 (0.700 to 0.735) | 0.50 (0.27 to 0.75) | 1.42 (1.26 to 1.59) |
| **Previous models** | | | |
| **Tyson 2008 [18]**<br>(N = 2,577) | 0.700 (0.664 - 0.710) | -0.36 (-0.46 to -0.26) | 0.76 (0.66 to 0.86) |
| **Manktelow 2013 [17]**<br>(N = 5,642) | 0.758 (0.741 - 0.774) | -0.84 (-0.92 to -0.77) | 0.91 (0.84 to 0.98) |
| **Santhakumaran 2018 [2] (N = 5,879)** | 0.754 (0.738 - 0.770) | -0.31 (-0.41 to -0.21) | 1.06 (0.98 to 1.15) |
| **NICHD 2020 [16]**<br>(N = 2,577) | 0.700 (0.667 - 0.712) | -0.35 (-0.45 to -0.24) | 0.82 (0.71 to 0.93) |

ROC denotes Receiver Operating Characteristic. Machine learning approaches are arranged in descending order of area under the ROC curve.

population-based real-world clinical data. Our tool is novel in providing information on how risk predictions are made for individual infants. The tool based on logistic regression demonstrates good model performance in the temporal internal validation cohort. The performance in the 'external validation' cohort was acceptable considering it was a cohort of infants from four different countries with strict eligibility criteria for clinical trial recruitment, and likely represents a slightly more restricted population with more immature and sicker infants than the whole extremely preterm population. Our tool also demonstrates a superior net benefit to previously published models [2,15–17], especially across the mortality threshold probabilities of between 10 and 70%. This threshold range is likely to represent the circumstances when decision-making and parental discussions are more challenging [13]. Survival-focused care, i.e., resuscitation and subsequent intensive care, is likely to be provided to infants with a very low mortality risk, whereas comfort care may be provided to those infants with an extremely high mortality risk and low chance of survival.

The logistic regression algorithm had similar overall model performance to the other machine learning approaches but offers a greater level of interpretability. The use of SHAP values to quantify predictors offers insights into the changing impact they have across the range of gestations. As expected, gestational age, birth weight z score and exposure to antenatal corticosteroids had the greatest impact on survival. Male sex has a negative effect on survival. Birth in a centre without an NICU, multiple pregnancy and prolonged rupture of membranes also negatively influence survival. The online tool allows users to explore the impact of modifiable factors, i.e., provision of antenatal corticosteroids and transfer to a centre with an NICU, to understand their influence on survival and so potentially support decision-making.

### Individualised prediction

Mortality is a key factor in perinatal decision-making [20]. The tool developed allows more individualised mortality prediction in the perinatal period, which is much needed in view of the complexity of the current neonatal care. Individualised prediction may provide an objective measure to address biases that may be inherent within healthcare professionals [21],

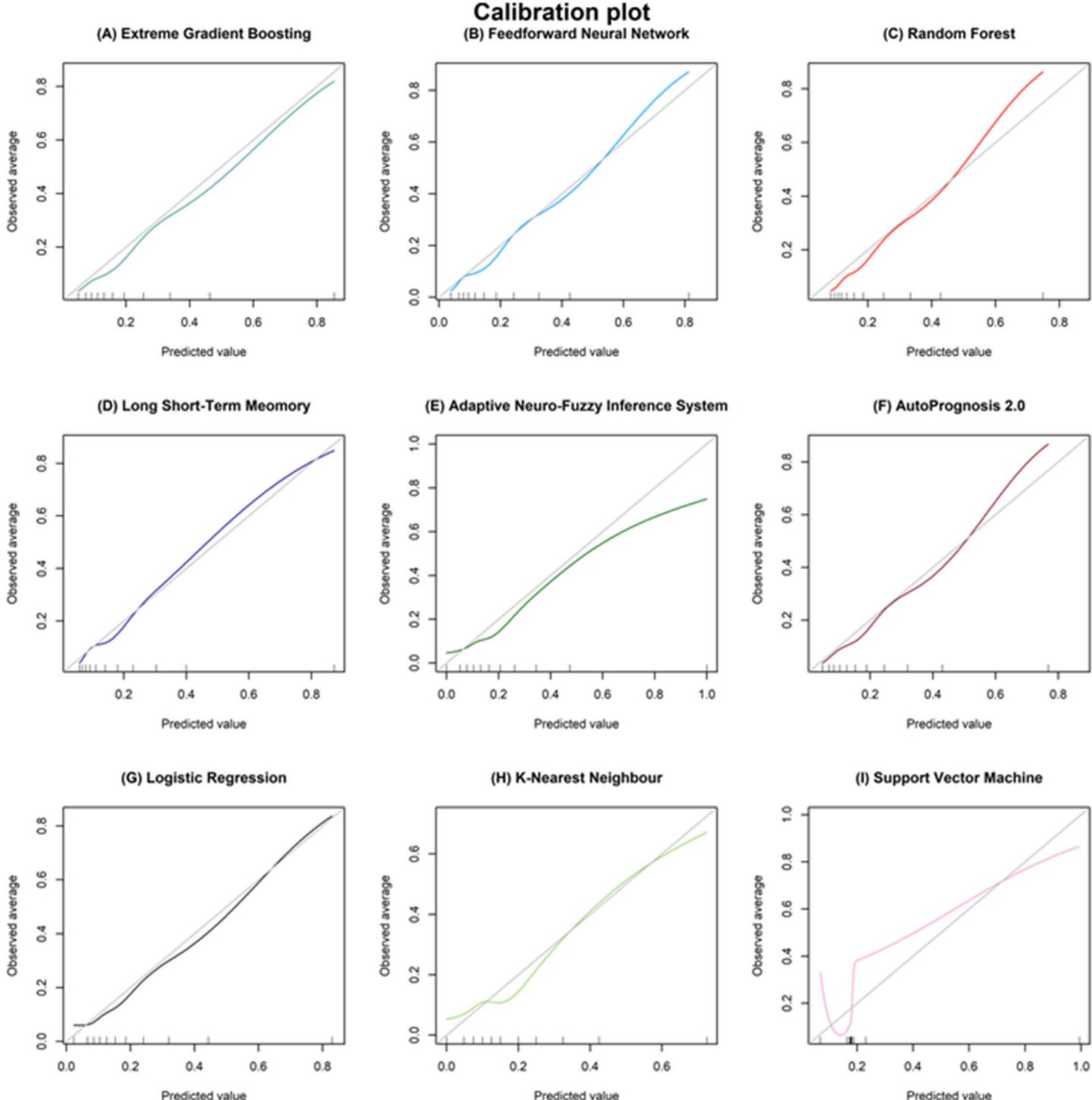

**Fig 2. Calibration plot of the machine learning approaches in the 'test' cohort (N=5,879).** Perfect calibration is seen when the predicted value (x-axis) matches the observed average of mortality (y-axis), i.e., when the machine learning approach follows the diagonal grey line.

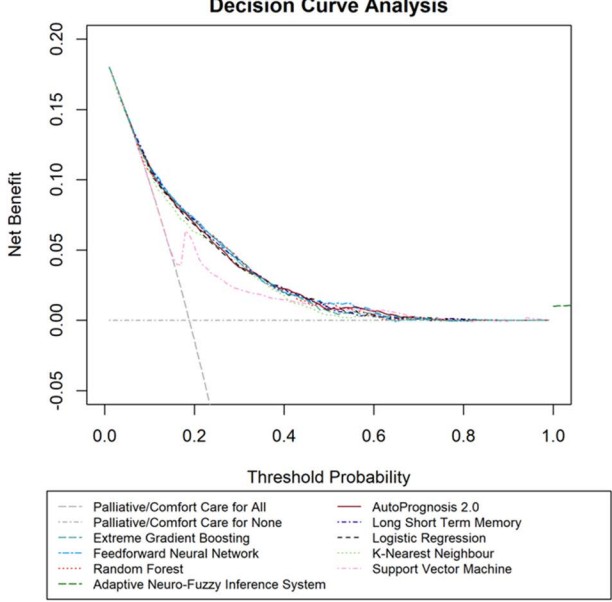

**Fig 3. Decision curve analysis of the machine learning approaches in the 'test' cohort (N = 5,879).** The threshold probability denotes the mortality risk threshold above which healthcare professionals would provide palliative/comfort care. Net benefit denotes the 'net' mortality cases detected per 100 infants. Net benefit is calculated as a weighted combination of true and false positives based on the threshold probability.

affecting how perinatal counselling is carried out. Current resources to support perinatal counselling provide estimates of survival based on broad criteria, especially gestational age groups in weeks [12,13]. However, mortality outcome varies depending on a range of factors, and it is unclear how these interact with one another to determine the mortality outcome. Furthermore, parents prefer a more individualised approach during perinatal counselling [7]. Our focus groups also demonstrated that understanding which factors contributed to the mortality risk would support parents in shared decision-making. As with all risk prediction models, the individualised predictions should be interpreted based on the clinical condition of the infant, with clinical input from healthcare professionals. It should be viewed as part of a package of supporting information that healthcare professionals have to hand to inform shared decision-making discussions, rather than making a clinical decision solely based on the output of the tool.

In addition to the support it could offer for decision-making, it could also have a role to play in research. Using the key characteristics which are important to the risk of infant death in this population could allow a more personalised approach to trial recruitment based on this risk, and could be enhanced with the addition of other multi-omic biomarkers [22]. This would be particularly useful for high-risk treatments by allowing identification of patients most likely to benefit, whilst at the same time avoiding exposing those least likely to benefit, but where the adverse effects may carry significant risk.

### Challenges/limitations

Our study attempted to address many of the common challenges [23] faced when developing and validating prediction tools, as detailed below.

**Quality of the dataset.** Data inaccuracies and missing data could not be controlled for as the data were entered at the point of care. However, predictors used in the dataset were derived from multiple data items within the dataset to ensure the validity of the variables extracted and reduce the missing data rate. At present, the dataset used to develop the tool does not include infants who died in the labour suite or obstetric theatre before admission to

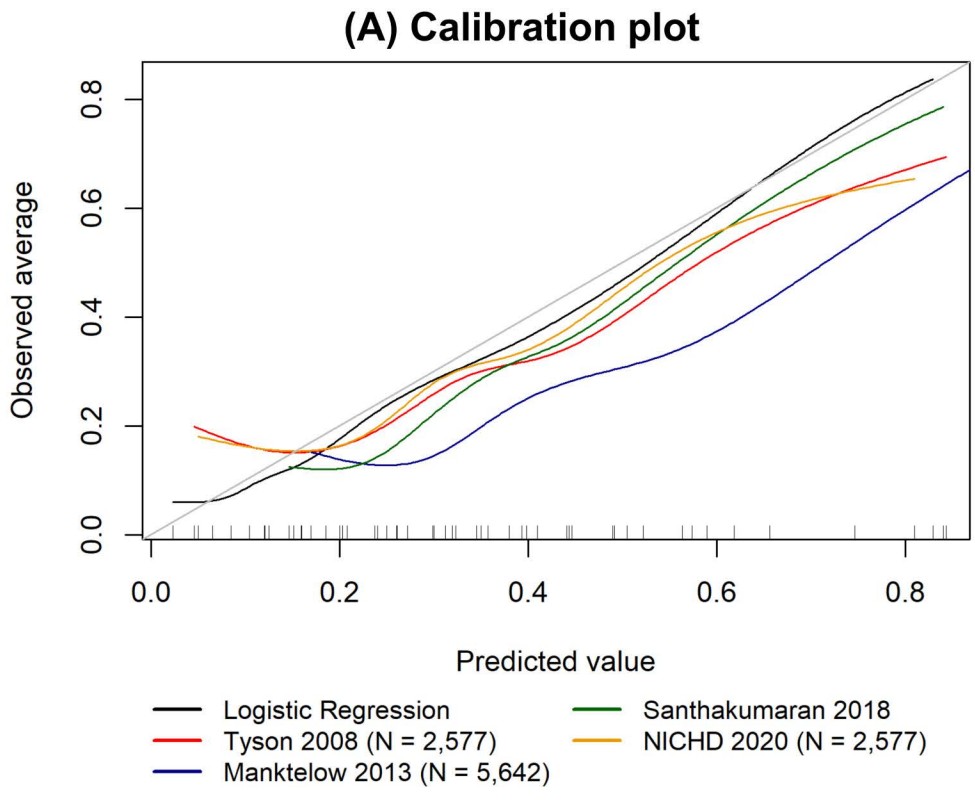

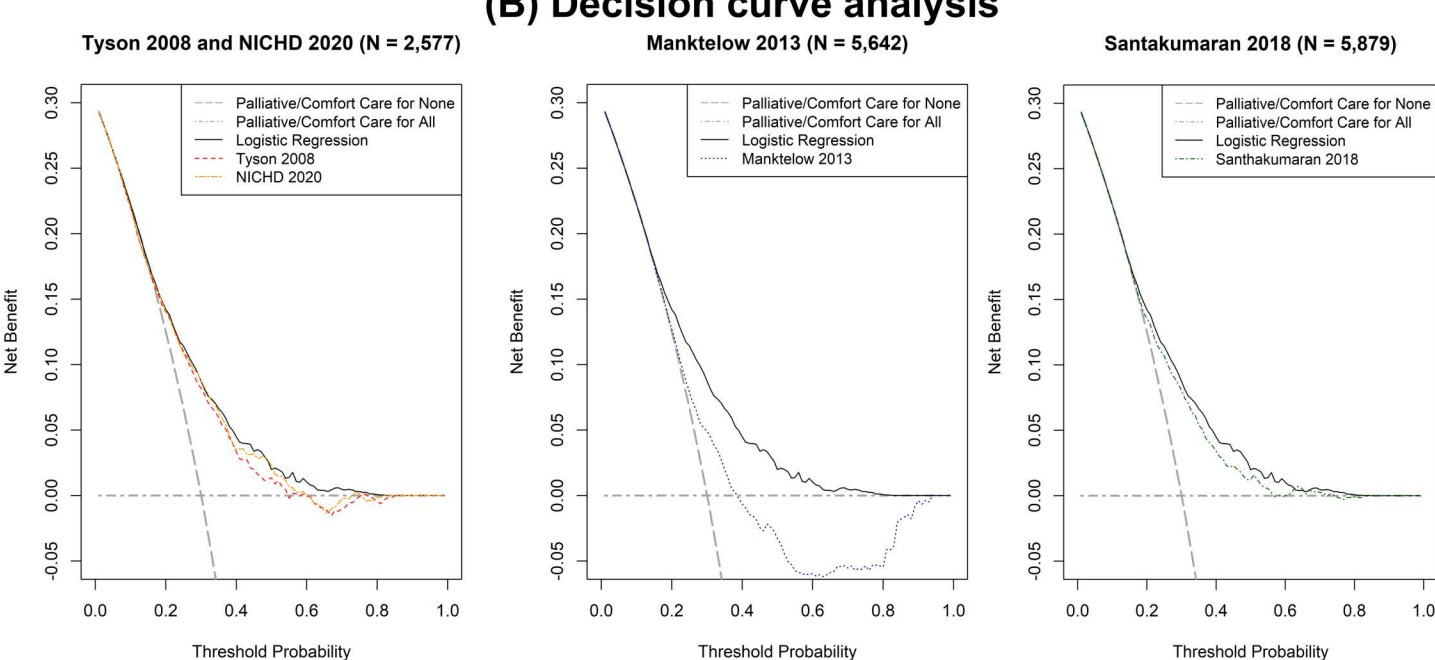

**Fig 4. A, Calibration plot and B, decision curve analysis comparing the logistic regression model with four previously published models in the 'test' cohort (N = 5,879).**

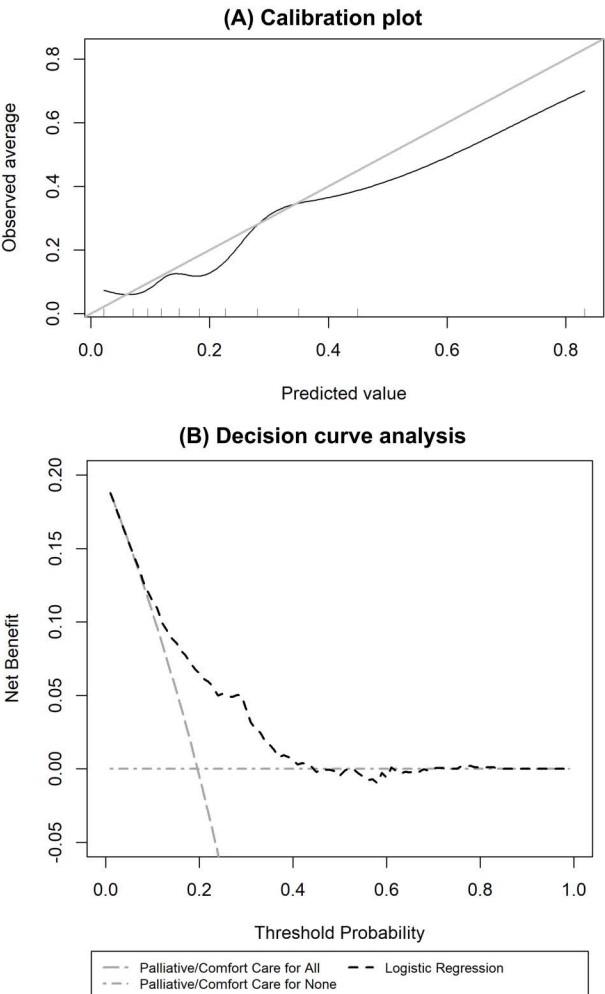

**Fig 5. A, Calibration plot and B, decision curve analysis of the developed logistic regression model in the 'external validation' cohort (N = 1,051).**

neonatal units. This may result in underestimating the true mortality risk among the highest-risk group of infants, typically those <24 weeks of gestation. We used birth weight z scores, which are not known prior to birth, rather than estimated fetal weight z scores, which may be quite different to the actual birth weight, potentially leading to inaccurate mortality estimates.

**Generalisability of the tool.** The dataset used to develop our tool encompasses all neonatal admissions of extremely preterm infants in England and Wales from 2012 and 2014, respectively. Hence, it reflected current perinatal practice with a diverse infant cohort representative of the national population. However, the variation in the model performance of our developed tool in different maternal ethnicities and neonatal networks must be interpreted with caution due to the smaller sample size and number of deaths within each subgroup, and the high missing data of 26% in maternal ethnicities. The variation in neonatal outcomes across different maternal ethnicities [24] and neonatal networks [25] may also partly explain some of the differences in model performance found. However, maternal ethnicity and region of care were not included as predictors, as our tool was intended to be used widely across high-income countries. Caution is needed when using the tool in the subgroups with poor performance. Predictions provided by the tool should be considered within the

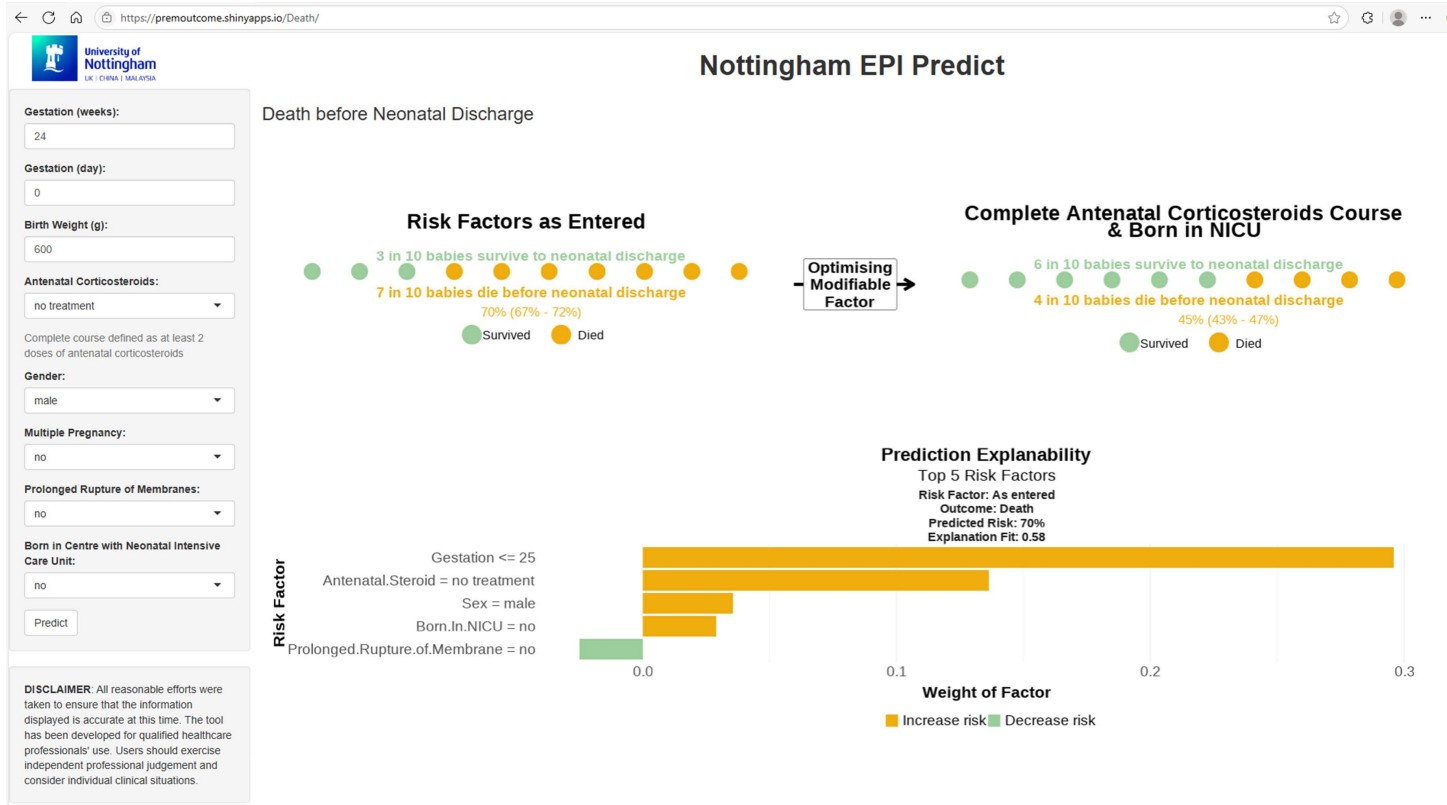

**Fig 6. Online prediction tool (**https://premoutcome.shinyapps.io/Death/**) based on the logistic regression model with explainability provided using Local Interpretable Model-agnostic Explanations (LIME).** In this example, the predicted risk (95% confidence interval) of mortality is 70% (67% to 72%) in a male infant born at $24^{+0}$ weeks' gestation, birth weight 600 grams, with no exposure to antenatal corticosteroids, and born in a centre with no co-located NICU. The predicted risk is driven by the gestation and no exposure to antenatal corticosteroids. The predicted risk decreases to 45% (43% to 47%) if the mother had received a complete course of antenatal corticosteroids (two doses of corticosteroids before birth) and given birth in a centre with a co-located NICU.

clinical context of the pregnancy and infant. Furthermore, the performance of our tool in different ethnicities and regions of care needs to be continuously monitored to avoid exacerbating ingrained inequalities in the current healthcare system.

**Computational expense.** Temporal internal validation was used as it was computationally expensive to analyse the entire dataset using a bootstrapping approach. Our models did not account for birth year as a predictor, as we anticipated that these would be accounted for by the changes in perinatal predictors across the birth years. Furthermore, our tool demonstrated good model performance in the temporal internal validation despite the changes in perinatal characteristics and neonatal outcomes with time.

**Perinatal predictors.** Mortality is predicted for the whole neonatal admission, but is based on early perinatal characteristics. The accuracy is therefore unlikely to significantly improve beyond the current routinely collected baseline data without the inclusion of important postnatal events during the NICU stay, and this likely explains why the different models show similar results. Furthermore, the majority of the predicted risks derived by the nine machine learning approaches were below 20% (S3 Table), despite good calibration for some of the approaches (Fig 2). This may be a reflection of the imbalanced dataset, with only 20% infant mortality in the 'test' cohort. The current models may also be too conservative due to a lack of variation within the perinatal predictors, and the distribution of predicted risk may improve with further postnatal clinical data.

Further development of the tool will explore the addition of dynamic postnatal clinical data and events, alongside multi-omics biomarkers. This will also allow prediction of other important morbidities, such as severe bronchopulmonary dysplasia or intraventricular haemorrhage, and neurodevelopmental outcomes, which are of prime importance to parents and are often discussed during the perinatal counselling process [13,21,26].

**Clinical implementation and explainability.** Despite our tool demonstrating promising findings, further external validation and impact studies on the usability and explanability of the tool, alongside its impact on real-world clinical decision-making, are needed before consideration for use in clinical practice to support shared clinical decision-making and to facilitate clinical implementation. The variable importance demonstrated by the tool for each individualised prediction provided a "technical" measure of explanability, rather than true explanability, as more work is needed to assess the tool from a causal reasoning or decision logic perspective. However, the visualisation of variable importance within the online tool, coupled with the co-design of the tool with focus groups of parents of extremely preterm infants, is a key first step taken in facilitating clinical implementation and explainability.

It is also unclear how our tool performs compared to clinical judgement alone in predicting mortality in extremely preterm infants. Previously published studies demonstrated the complexity and variation of mortality risk assessment in extremely preterm infants among perinatal healthcare professionals. Healthcare professionals were found to underestimate [27,28] or overestimate [29,30] the mortality risk, demonstrating a need for an objective measure in risk assessment and consideration of care plans for extremely preterm infants.

## Conclusion

Our individualised, online mortality prediction tool demonstrated the potential of applying machine learning approaches to population-based routinely recorded electronic patient data to predict neonatal outcomes in preterm infants. Further external validation and impact analysis studies are needed to understand if this could support parental discussion and shared perinatal clinical decision-making.

## Materials and methods

### Ethics statement

Ethical approval was granted by the Sheffield Research Ethics Committee (REC reference 19/YH/0115). The study was reported using the Transparent Reporting of a multivariable prediction model for Individual Prognosis Or Diagnosis AI (TRIPOD+AI) checklist [31] and the Transparency, Reproducibility, Ethics and Effectiveness (TREE) framework [32]. The PLUSS Trial Steering Committee approved the use of grouped trial data for the external validation.

### Study population and data source

**Model development and internal validation.** Routinely entered electronic patient record data from a retrospective population-based cohort of extremely preterm infants born between $23^{+0}$ and $27^{+6}$ weeks' gestation and admitted to all 185 neonatal units in England and Wales between January 2010 and December 2020 were extracted from the National Neonatal Research Database (NNRD) [33]. This represented over 90% of neonatal units in England in 2010 with complete coverage in England from 2012 and Wales from 2014. Infants with birth weight z score < -4 or >4 based on the UK WHO preterm growth reference [34] or discharged to non-participating neonatal units were excluded as these were potentially erroneous or incomplete entries.

The study population was split into three cohorts depending on the birth year of the infants. Firstly, infants born from 2010 to 2015 ('training' cohort) were used to develop the mortality prediction models using different hyperparameters for each of the nine machine learning approaches. Secondly, infants born from 2016 to 2017 ('validation' cohort) were used to identify the optimal hyperparameters that produced the best discrimination performance for each of the machine learning approaches. Lastly, infants born from 2018 to 2020 ('test' cohort) were used to internally validate the performance of each

of the machine learning approaches using the optimal hyperparameters identified previously [35]. The 'test' cohort was also used to compare the performance of our preferred final model implemented in our developed online mortality prediction tool with previously published models [2,15–17].

**External validation.** The developed prediction tool was externally validated on infants born between $23^{+0}$ and $27^{+6}$ weeks' gestation enrolled in the PLUSS trial [19] from 21 NICUs in four countries (Australia, New Zealand, Singapore and Canada) from January 2018 to March 2023. Infants were recruited in the first 48 hours of age if they were either invasively ventilated or receiving non-invasive ventilation with a decision to treat with surfactant.

## Predictors

Nine perinatal predictors were determined *a-priori* based on their clinical significance and association with mortality from the literature review [36,37], framework of practice [13] and national guidance [12]. No postnatal predictors nor biomarkers were included.

The predictors recorded at or shortly after delivery were: (i) gestational age determined using the best obstetric estimate [38]; (ii) birth weight z-score as a measure of fetal growth; (iii) sex; (iv) multiple pregnancy; (v) exposure to complete (defined as at least two doses of ANC) or incomplete ANC; (vi) prolonged rupture of membranes of more than 18 hours; (vii) chorioamnionitis; (viii) major congenital anomaly based on the European Surveillance of Congenital Anomalies (EUROCAT) registry [39]; and (ix) born in a maternity centre with a co-located Level 3 NICU [40]. Definitions of these predictors are described in S6 Table. Due to the retrospective nature of the study, the assessment of the predictors and outcome was not blinded.

## Outcome

Mortality was defined as death of the infant before discharge from neonatal units.

## Statistical analysis

The extracted dataset was cleaned using STATA 15 [41].

**Machine learning approaches.** The prediction models were developed in the 'training and validation' cohort using R version 4.3.2 in Rstudio [42] and Python version 3.11.7 in Jupyter Notebook [43] for nine machine learning approaches. The nine approaches were Logistic Regression, AutoPrognosis 2.0, Adaptive Neuro-Fuzzy Inference System, Extreme Gradient Boosting, Feedforward Neural Network, K-Nearest Neighbour, Long Short-Term Memory, Random Forest and Radial Kernel Support Vector Machine (Table 2). For the latter seven machine learning approaches, the dataset was z-score normalised based on the training data values to ensure stability and speed up the convergence of the algorithms. A complete case analysis was used for all approaches, excluding infants with missing data for the predictor variables. Further details of the modelling approaches are described in S1 File. For each of the nine machine learning approaches, the importance of each predictor to the prediction in the 'test' cohort was determined using the mean SHAP values. These were derived from Kernel SHAP [44] using the kernelshap package in R [45] and a sample of 100 infants to calculate the marginal expectation. As a subgroup analysis, the importance of the predictors to the prediction of the best-performing algorithm was determined for each gestational week in the 'test' cohort using mean SHAP values.

**Model performance.** Performance of the nine machine learning approaches was assessed by temporal internal validation in the 'test' cohort across the three domains of (i) discrimination (AUROC) [46], (ii) calibration (calibration plot, calibration-in-the-large and calibration slope) [47], and (iii) utility measures (decision curve analysis) [48]. Decision curve analysis is a decision-analytic measure to summarise the 'net' benefit of the model in detecting neonatal death across a range of mortality risk thresholds above which healthcare professionals would provide palliative/comfort care (threshold probability).

The performance of the previously developed mortality prediction models (Tyson 2008 [18], Manktelow 2013 [17], San-thakumaran 2018 [2] and NICHD 2020 [16]) was also assessed in the 'test' cohort using published regression coefficients for the first three models and the freely available online NICHD 2020 [16] tool. Only infants within the 'test' cohort who fulfilled the inclusion criteria for each of the previously published models were used to assess their performance. Sub-group analysis was also performed to assess the performance of the best-performing algorithm across different maternal ethnicities and regions where perinatal care was provided (neonatal networks). The performance of the best-performing algorithm was then externally validated on the PLUSS trial [19] cohort of infants.

**Online prediction tool.** An online prediction tool was developed based on the best-performing algorithm using the shiny package in R [49] and deployed on shinyapps.io. The tool displays the predicted mortality risk for each infant alongside the predicted mortality risk if the modifiable predictors of exposure to antenatal corticosteroids and being born in a centre with a co-located NICU were optimised. Explainability of the individual predictions made was provided using the Local Interpretable Model-Agnostic Explanation (LIME) package in R [50]. The 'training and validation' cohort was used to train the LIME explainer. Once the LIME explainer is created, 5,000 permutations using forward selection in a ridge regression model and Gower's distance were used to explain how the individual predictions were made. The tool was co-designed with parents of ex-preterm infants through two focus group sessions organised with the Spoons charity.

## Supporting information

**S1 Table. Table demonstrating the characteristics of infants with complete data included in developing and validating the prediction model, and excluded infants with missing data for birth weight z score, multiple pregnancy, exposure to antenatal corticosteroids or born in a centre with neonatal intensive care unit (NICU).** IQR denotes interquartile range.
(DOCX)

**S1 File. File detailing the model development process for each of the nine machine learning approaches.**
(DOCX)

**S2 Table. Table demonstrating the importance of the predictors to the predictions in the testing dataset based on mean SHapley Additive exPlanations (SHAP) values.** NICU denotes Neonatal Intensive Care Unit. PROM denotes Prolonged rupture of membranes. [1] Chorioamnionitis and congenital anomalies predictors were dropped from the logistic regression model after backward stepwise selection. [2] Sex and Multiple pregnancy predictors were used to group the dataset in the Adaptive Neuro-Fuzzy Inference System.
(DOCX)

**S1 Fig. Figure demonstrating the importance of the predictors to the predictions of the logistic regression model in the 'test' cohort based on mean SHapley Additive exPlanations (SHAP) values for the overall cohort and by gestational week groups.**
(DOCX)

**S3 Table. Table describing the distribution of the predicted risk obtained from the nine machine learning approaches used in the 'test' cohort (N = 5,879).**
(DOCX)

**S4 Table. Table presenting the regression coefficients with 95% confidence intervals (CI) of the final logistic regression model to predict death before neonatal discharge.** Regression coefficient denotes estimated change in the log odds of death when the associated predictor increases by one unit or in comparison with a reference category. NICU denotes Neonatal Intensive Care Unit.
(DOCX)

**S5 Table. Table describing the incidence of death and the model performance of the logistic regression approach in the 'test' cohort (N = 5,879) stratified by (A) maternal ethnicity (N = 4,379) and (B) neonatal network (N = 5,824).** ROC = Receiver Operating Characteristic. 1,500 (26%) and 55 (1%) of infants with missing data on maternal ethnicity and neonatal network were excluded.
(DOCX)

**S2 Fig. Calibration plot of the logistic regression approach in the 'test' cohort (N = 5,879) stratified by (A) maternal ethnicity (N = 4,379) and (B) neonatal network (N = 5,824).**
(DOCX)

**S3 Fig. Decision curve analysis of the logistic regression approach in the 'test' cohort (N = 5,879) stratified by (A) maternal ethnicity (N = 4,379) and (B) neonatal network (N = 5,824).**
(DOCX)

**S6 Table. Table describing the definition of variables extracted from the National Neonatal Research Database.**
(DOCX)

**S7 Table. Participating neonatal units in England and Wales and their respective lead clinicians.** The list was accessed from https://www.imperial.ac.uk/neonatal-data-analysis-unit/neonatal-data-analysis-unit/list-of-national-neonatal-units/ on 06/01/2022.
(DOCX)

## Acknowledgments

Electronic patient data recorded at participating neonatal units are transmitted to the Neonatal Data Analysis Unit (NDAU) to form the NNRD (S7 Table). We are grateful to all the families that agreed to include their infant's data in the NNRD, the health professionals who recorded data and the NDAU team. We are also grateful to the Spoons charity and all parents who participated in the focus group sessions in co-designing the online tool. We would also like to thank all the families and infants who participated in the PLUSS trial and the health professionals who cared for them during their neonatal stay. We would like to acknowledge the significant contribution of the PLUSS trial investigators (Omar F. Kamlin, Jeanie L. Y. Cheong, Jennifer A. Dawson, Susan E. Jacobs, Lex W. Doyle, Peter G. Davis (Royal Women's Hospital, Melbourne, Australia), Susan M. Donath (Murdoch Children's Research Institute, Melbourne, Australia), Peter A. Dargaville (Menzies Institute for Medical Research, Hobart, Australia), Pita Birch (Mater Mother's Hospitals, Brisbane, Australia), Steven M. Resnick (King Edward Memorial Hospital, Perth, Australia), Georg M. Schmölzer, Brenda Law (Centre for the Studies of Asphyxia and Resuscitation, Edmonton, Canada), Risha Bhatia (MonashChildren'sHospital, Melbourne, Australia), Katinka P. Bach (Newborn Services, Starship Child Health, Auckland, New Zealand), Koertde Waal, Javeed N. Travadi (John Hunter Children's Hospital, Newcastle, Australia), Pieter J. Koorts (Royal Brisbane and Women's Hospital, Brisbane, Australia), Mary J. Berry (University of Otago, Wellington, New Zealand), Kei Lui (Royal Hospital for Women, New South Wales, Australia), Victor S. Rajadurai, Suresh Chandran (KK Women's and Children's Hospital, Singapore) Martin Kluckow (Royal North Shore Hospital, New South Wales, Australia), Elza Cloete (Christchurch Women's Hospital, Christchurch, New Zealand), Margaret M. Broom (Canberra Hospital, Canberra, Australian), Michael J. Stark (The Women's and Children's Hospital, Adelaide, Australia), Adrienne Gordon (Royal Prince Alfred Hospital, New South Wales, Australia), Vinayak Kodur (Te Whatu Ora Waikato, Hamilton, New Zealand)).

## Author contributions

**Conceptualization:** T'ng Chang Kwok, Don Sharkey.

**Data curation:** T'ng Chang Kwok, Kate L. Francis, Christopher J. D. McKinlay, Brett J. Manley.

**Formal analysis:** T'ng Chang Kwok, Chao Chen, Jayaprakash Veeravalli, Carol A.C. Coupland, Edmund Juszczak, Jonathan Garibaldi, Kate L. Francis.

**Funding acquisition:** T'ng Chang Kwok, Carol A.C. Coupland, Edmund Juszczak, Jonathan Garibaldi, Christopher J. D. McKinlay, Brett J. Manley, Don Sharkey.

**Investigation:** T'ng Chang Kwok, Chao Chen, Jayaprakash Veeravalli, Carol A.C. Coupland, Edmund Juszczak, Jonathan Garibaldi, Kirsten Mitchell, Kate L. Francis, Don Sharkey.

**Methodology:** T'ng Chang Kwok, Chao Chen, Jayaprakash Veeravalli, Carol A.C. Coupland, Edmund Juszczak, Jonathan Garibaldi, Kirsten Mitchell, Kate L. Francis, Don Sharkey.

**Project administration:** T'ng Chang Kwok, Kirsten Mitchell.

**Resources:** T'ng Chang Kwok.

**Software:** T'ng Chang Kwok.

**Supervision:** Carol A.C. Coupland, Edmund Juszczak, Jonathan Garibaldi, Kirsten Mitchell, Don Sharkey.

**Validation:** T'ng Chang Kwok, Chao Chen, Carol A.C. Coupland, Edmund Juszczak, Jonathan Garibaldi.

**Visualization:** T'ng Chang Kwok.

**Writing – original draft:** T'ng Chang Kwok, Don Sharkey.

**Writing – review & editing:** T'ng Chang Kwok, Chao Chen, Jayaprakash Veeravalli, Carol A.C. Coupland, Edmund Juszczak, Jonathan Garibaldi, Kirsten Mitchell, Kate L. Francis, Christopher J. D. McKinlay, Brett J. Manley, Don Sharkey.

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
