## [Decision Letter · Decision Letter 0]

26 Aug 2025

Developing and validating an explainable digital mortality prediction tool for extremely preterm infants

PLOS Digital Health

Dear Dr. Sharkey,

Thank you for submitting your manuscript to PLOS Digital Health. After careful consideration, we feel that it has merit but does not fully meet PLOS Digital Health's publication criteria as it currently stands. Therefore, we invite you to submit a revised version of the manuscript that addresses the points raised during the review process.

Please submit your revised manuscript within 60 days Oct 25 2025 11:59PM. If you will need more time than this to complete your revisions, please reply to this message or contact the journal office at digitalhealth@plos.org. Please include the following items when submitting your revised manuscript:

* A rebuttal letter that responds to each point raised by the editor and reviewer(s). You should upload this letter as a separate file labeled 'Response to Reviewers '. This file does not need to include responses to any formatting updates and technical items listed in the 'Journal Requirements' section below.

* A marked-up copy of your manuscript that highlights changes made to the original version. You should upload this as a separate file labeled 'Revised Manuscript with Track Changes '.

* An unmarked version of your revised paper without tracked changes. You should upload this as a separate file labeled 'Manuscript '.

We look forward to receiving your revised manuscript.

Kind regards,

Chenxi Yang, Ph.D.

Academic Editor

PLOS Digital Health

Leo Anthony Celi

Editor-in-Chief

PLOS Digital Health

orcid.org/0000-0001-6712-6626

**Journal Requirements:**

1. Please update your online Competing Interests statement. If you have no competing interests to declare, please state: “The authors have declared that no competing interests exist.”

2. In this instance it seems there may be acceptable restrictions in place that prevent the public sharing of your minimal data. However, in line with our goal of ensuring long-term data availability to all interested researchers, PLOS’ Data Policy states that authors cannot be the sole named individuals responsible for ensuring data access (http://journals.plos.org/digitalhealth/s/data-availability#loc-acceptable-data-sharing-methods).

3. Please provide separate main figure files in .tif or .eps format only and ensure that all files are under our size limit of 10MB. You may leave the embedded figures in the manuscript

For more information about how to convert your figure files please see our guidelines: https://journals.plos.org/digitalhealth/s/figures

4. Some material included in your submission may be copyrighted. According to PLOS’s copyright policy, authors who use figures or other material (e.g., graphics, clipart, maps) from another author or copyright holder must demonstrate or obtain permission to publish this material under the Creative Commons Attribution 4.0 International (CC BY 4.0) License used by PLOS journals. Please closely review the details of PLOS’s copyright requirements here: PLOS Licenses and Copyright. If you need to request permissions from a copyright holder, you may use PLOS's Copyright Content Permission form.

Potential Copyright Issues:

Figure 6 contains branding/a logo. We are not permitted to publish this under our CC-BY 4.0 license, even with permission. We ask that you please remove or replace it.

**Additional Editor Comments (if provided):**
**Reviewers' Comments:**

Reviewer #1: The current manuscript does not clearly articulate the significance, objectives, and clinical implications of mortality prediction. While the authors mention risk factors and existing prediction models, there is no systematic discussion of why individualized mortality prediction is needed or how such predictions can provide practical value for family counseling and clinical practice. Moreover, the main technical challenges related to methodology, data, and clinical application are not clearly summarized. I recommend adding relevant content to help readers better understand the innovation and real-world significance of this research.

The authors state in the Introduction that a major limitation of current mortality prediction models for extremely preterm infants is their lack of actual clinical implementation, which is a key challenge in the field. However, although this study has developed and validated a new prediction tool, it does not evaluate its deployment, user feedback, or support for real-world clinical decision-making. Thus, while there are advances in modeling and data scale, the work does not address the critical bottleneck of clinical translation. I suggest the authors clarify this limitation in the Discussion and provide more concrete recommendations for future efforts to facilitate clinical implementation.

The use of SHAP and LIME to illustrate feature contributions is primarily limited to presenting the statistical importance of input variables. It should be noted that such 'explainability' is confined to variable ranking at the technical level and does not address the causal reasoning, decision logic, or medical context that clinicians require in practice. Therefore, the model’s explainability remains largely technical and falls short of true clinical interpretability and trustworthiness. I recommend that the authors expand their discussion of the current state and limitations of model explainability, especially in relation to the real needs of clinicians and families, and explore how to enhance clinical utility and interpretability in future work.

The potential negative impacts of model deployment, such as families becoming overly reliant on model predictions, resulting in pessimistic or inappropriate decision-making, or the risk of doctor-patient disputes and inequitable resource allocation, are not sufficiently addressed. The manuscript only briefly mentions the need for further evaluation, lacking substantive discussion or risk warnings.

The modeling process is described in very general terms; while nine machine learning methods are compared, the specific implementation details, parameter settings, and feature engineering procedures are not disclosed. There is no clear description of model training, validation, or tuning processes, including how the training/validation/test sets were divided, the cross-validation strategies used, whether tuning was automated, or whether measures such as early stopping and regularization were employed.

While the authors mention the use of complete case analysis for missing data, there is no information on the missingness rates for each variable, or on whether sensitivity analyses or imputation were performed.

it's recommend providing analysis code or a detailed modeling protocol as supplementary material to enhance the study’s reproducibility.

The results show that model calibration or performance declines in certain subgroups (e.g., infants of Black mothers, certain regions). The authors should further analyze the reasons and potential impact, and discuss possible improvement strategies, with appropriate additions to the Discussion.

It should be explicitly stated that this model does not include infants who died before NICU admission, which may result in underestimating mortality rates among the highest-risk population. I also recommend discussing the limitations of using only perinatal variables and the potential for incorporating dynamic clinical data, postnatal complications, and multi-omics biomarkers in future models.

Reviewer #2: This study addresses the important and challenging task of mortality prediction in extremely preterm infants using routinely collected electronic health record data. The authors compared nine machine learning approaches and developed an explainable tool based on stepwise backward logistic regression.The topic is clinically relevant and the study design is rigorous; however, some methodological clarifications and additional analyses are needed to strengthen the manuscript.

Major Weakness:

1. Please provide a clear description of how each input feature was numerically encoded. For example, specify that Sex was coded as 1 (Male) and 0 (Female). Such details will help readers interpret the SHAP plots more accurately.

2. The font size of the feature names and the numeric values on the x-axis in the SHAP plots should be increased or otherwise adjusted to improve clarity and readability.

3. In Table 2, the values for Calibration-in-the-large and Calibration slope are presented. Please explain how these metrics were derived in the methods section so that readers can better understand their calculation and interpretation.

4. According to the calibration plot, overfitting does not appear to be a major issue across the machine learning methods tested. However, for traditional machine learning methods, calibration analysis may be of limited relevance since these approaches are inherently less prone to overfitting. Given that the AUROC values are only around 0.75, I encourage the authors to provide the distribution of predicted risks in the test set (e.g., the number of cases with predicted probability >0.8, 0.6–0.8, 0.4–0.6, etc.). This will help readers assess the practical utility of the models in real-world decision-making. If the number of high-confidence predictions (e.g., >0.8) is very small and most predictions cluster around 0.5, the limitations of the approach should be discussed more explicitly.

Reviewer #3: This manuscript describes the development and validation of an explainable mortality prediction tool for preterm infants using a large dataset, and an external validation cohort. The authors used and compared nine machine learning approaches and chose a logistic regression model for its interpretability and comparable performance. An online tool is made available for clinical use.

Please find the specific comments as attached.

**Figure resubmission:**

**Reproducibility:** To enhance the reproducibility of your results, we recommend that authors of applicable studies deposit laboratory protocols in protocols.io, where a protocol can be assigned its own identifier (DOI) such that it can be cited independently in the future. Additionally, PLOS ONE offers an option to publish peer-reviewed clinical study protocols. Read more information on sharing protocols at https://plos.org/protocols?utm_medium=editorial-email&utm_source=authorletters&utm_campaign=protocols

---

## [Decision Letter · Decision Letter 1]

15 Nov 2025

Developing and validating an explainable digital mortality prediction tool for extremely preterm infants

PDIG-D-25-00440R1

Dear Professor Sharkey,

We are pleased to inform you that your manuscript 'Developing and validating an explainable digital mortality prediction tool for extremely preterm infants' has been provisionally accepted for publication in PLOS Digital Health.

Best regards,

Chenxi Yang, Ph.D.

Academic Editor

PLOS Digital Health

**Additional Editor Comments (if provided):**

**Reviewer Comments (if any, and for reference):**

All comments have been addressed